# Early Physiotherapy Intervention Program for Preterm Infants and Parents: A Randomized, Single-Blind Clinical Trial

**DOI:** 10.3390/children9060895

**Published:** 2022-06-15

**Authors:** Mirari Ochandorena-Acha, Marc Terradas-Monllor, Laura López Sala, Maria Engracia Cazorla Sánchez, Montserrat Fornaguera Marti, Isabel Muñoz Pérez, Thais Agut-Quijano, Martín Iriondo, Joan Carles Casas-Baroy

**Affiliations:** 1Research Group on Methodology, Methods, Models and Outcomes of Health and Social Sciences (M_3_O), Faculty of Health Sciences and Welfare, Centre for Health and Social Care Research (CESS), University of Vic-Central University of Catalonia (UVIC-UCC), C.Sagrada Família, 7, 08500 Vic, Barcelona, Spain; mirari.ochandorena@uvic.cat (M.O.-A.); joancarles.casas@uvic.cat (J.C.C.-B.); 2Sant Joan de Deu Barcelona Children’s Hospital, Passeig de Sant Joan de Déu, 2, 08950 Esplugues de Llobregat, Barcelona, Spain; laura.lopezs@sjd.es (L.L.S.); mariaengracia.cazorla@sjd.es (M.E.C.S.); montserrat.fornaguera@sjd.es (M.F.M.); isabel.munoz.perez@gmail.com (I.M.P.); thais.agut@sjd.es (T.A.-Q.); martin.iriondo@sjd.es (M.I.); 3Pain Medicine Section, Anaesthesiology Department, Hospital Clínic de Barcelona, 08036 Barcelona, Catalonia, Spain

**Keywords:** neonates, parent training, motor development, quantitative methods

## Abstract

Background: The early developmental interventions might be designed with a preventative approach to improving the development of at-risk preterm infants. The present study aimed to evaluate the effectiveness of an early physiotherapy intervention on preterm infants’ motor and global development, and on parents’ stress index. Methods: 48 infants were enrolled and randomized into two groups. Infants allocated to the intervention group received an early physiotherapy intervention, based on parental education sessions and tactile and kinesthetic stimulation during the NICU period, as well as a home-based activity program. The intervention commenced after 32 weeks post-menstrual age and ended at 2 months corrected age. Infants allocated to the control group received the usual care based on the NIDCAP-care. Results: No differences were found between groups on the Alberta Infant Motor Scale at 2- or 8-months corrected age. Infants in the intervention group showed more optimal fine motor, problem-solving, personal-social, and communication development at 1 month corrected age. Conclusions: The results showed no effect on the early physiotherapy intervention. Results might be related to the dose or intensity of the intervention, but also to the poor parental compliance. ClinicalTrials.gov NCT03313427.

## 1. Introduction

The knowledge of early human brain development has evidenced that the high developmental activity in the brain during the second half of gestation induces an increased vulnerability in infants born preterm [1,2]. Consequently, within their immaturity condition, preterm infants usually show a delay in their motor development. Compared to infants born at term age, premature infants often present lower muscle tone, less movement variability, and higher behavioral complexity. These characteristics compromise their postural control, which is necessary to gain stability, orientation, and spatial organization [3,4,5]. For example, when preterm infants attempt to lift their legs, they are often unable to keep the limbs off the support surface. In addition, when they succeed in raising their legs, they frequently roll to the side, demonstrating an inability to turn their head from side to side or lift one leg at a time [6,7]. Although these motor alterations are sometimes subtle, they might contribute to delays in other domains such as cognitive development, and later, academic performance [7,8]. Furthermore, the motor experiences contribute to the infants’ attempts and motivation to explore the environment. Their motor ability allows them to receive and interpret relevant information, as well as to communicate and interact physically with people and objects, and solve problems [8,9]. 

As a consequence of an unexpected delivery or the admission to the Neonatal Intensive Care Unit (NICU) [10], preterm infants’ parents usually experience high levels of anxiety, stress, or depression, and that, in turn, might influence children’s development [11,12]. Therefore, the infants’ and families’ quality of life might also be affected [7,8]. Previous research has shown that parents’ involvement in early interventions enables them to better understand their child’s development and learn how to support their child. Moreover, when parents provide hands-on programs, they experience a sense of autonomy, empowerment, and stronger bonding with their infant [12,13,14,15,16]. This large body of research has provided valuable information when designing and administrating developmental programs to enhance preterm infants’ development and improve their parents’ mental health [17,18]. 

Early developmental interventions are undertaken to improve brain connections during critical periods of infants’ central nervous system development [17]. Although some early interventions are provided only during the NICU period or at home [19,20], it seems that the duration is not long enough to offset the adverse outcomes of preterm birth [12,17]. Consequently, previous research concluded that interventions that commence at the NICU and continue after discharge are the most recommended for families. Furthermore, these interventions have been shown to have a more significant effect on infants’ outcomes [12,17,18,21]. 

NICU multidisciplinary teams are typically involved in assessing infants born preterm and are encouraged to provide early developmental interventions to the infants and their parents. Within the scope of physiotherapy, preventative, developmental interventions are targeted to at-risk preterm infants—without risk factors for adverse neurological outcomes in the perinatal period, such as intraventricular hemorrhage, periventricular leukomalacia, sensory abnormalities, specific motor abnormalities, or chromosomal abnormalities [10,12,17,21,22,23]. Preventative early interventions usually involve parents and aim to enhance the parent–infant relationship, improve parent coping and the home environment through parental education, and promote infants’ stimulation and autonomy [21]. Although many types of research have been conducted on early preventative and developmental interventions [24,25,26,27,28,29,30,31,32], there is a shortage of high-level evidence-based research within the scope of physiotherapy. 

This research aimed to evaluate the effectiveness of a preventative early physiotherapy intervention to improve at-risk preterm infants’ motor development at 2- and 8-months corrected age. The study also aimed to evaluate the intervention’s effectiveness in enhancing the global development of infants born preterm at 1 month corrected age and its post-intervention effects in decreasing parents’ stress. This study will consider post-intervention at infants’ 1- and 2-months corrected age, and short-term at 8 months corrected age [33]. The present study might contribute new knowledge to the physiotherapy field as part of the NICU multidisciplinary team involved in preterm infants and their parents’ care. We hypothesized that infants who receive early physiotherapy intervention show more significant motor performance improvements at 2- and 8-months corrected age compared to the infants in the control group. We also hypothesized that the intervention positively affects infants’ global development at 1 month corrected age and parents’ post-intervention stress level. 

## 2. Materials and Methods

### 2.1. Study Design

A randomized, single-blind clinical trial was conducted at the *** Hospital (***), a level-III hospital with NICU facilities. The study results have been reported according to the CONSORT 2010 Statement [34]. Ethical permission was obtained from the relevant Clinical Research Ethical Committee of *** (***). The protocol accepted by the Ethical Committee is registered at ClinicalTrials.gov (NCT***). All parents provided informed written consent. The study has been conducted according to Good Clinical Practice and Declaration of Helsinki principles [35].

### 2.2. Participants

Participants’ recruitment took place between December 2017 and December 2018. Preterm infants born between 28 + 0 and 34 + 0 (weeks + days) of gestational age and whose parents stayed at the hospital for more than 6 h a day and who were able to speak and understand Spanish were eligible for the study. Exclusion criteria were preterm triplets, major central nervous system injury (grade III/IV intraventricular hemorrhage or periventricular leukomalacia), severe musculoskeletal or congenital abnormalities, bronchopulmonary dysplasia, major surgery, severe sepsis or necrotizing enterocolitis during the neonatal period, hearing impairment or retinopathy of prematurity, and infants born of mothers with a documented history of social problems or mental illness. 

### 2.3. Sample Size

The sample size was calculated based on the primary outcome, the Alberta Infant Motor Scale (AIMS) [36]. A difference of 1.2 SD between groups was considered to be clinically significant [37]. Assuming a 10% drop-out rate, 24 infants in each group were required to ensure a statistical power of 80% chance of detecting this difference at a significance level of 0.05 (α) [38]. 

### 2.4. Randomization and Masking

Selected participants were recruited by a physical therapist (PT) after 32 + 0 weeks post-menstrual age (PMA) and only if the preterm children were medically stable (no need for invasive mechanical ventilation and no active sepsis). In the case of infants born after 32 weeks PMA, the recruitment took place 2–3 days after birth, when the infants were also medically stable. 

The random assignment of each preterm infant with his/her parents was stratified according to the gestational age at birth (between 28 + 0 and 31 + 6 and between 32 + 0 and 34 + 0 gestational age) to ensure comparability among groups. The stratification was accomplished using two computer-generated lists at a 1:1 ratio. Twins were assigned to the same group. The allocation was made through sealed opaque envelopes (identified according to stratification group and numbered consecutively), which were opened by the first author before parents were given oral and written information. Parents who agreed to participate in the study and signed informed consent were included in the study. 

Due to the nature of the intervention, neither the PTs conducting the intervention nor the participants (parents) were blinded to the group allocation. Both the PTs administering the postintervention assessments and nurses at the neonatal unit were blinded to the participants’ assigned group. The regular (non-researcher) nursing staff and PTs at the hospital were not taught about the experimental intervention, and they did not change their procedures or support provided to parents during the intervention period.

### 2.5. Intervention

**Standard care**. The hospital provides the Newborn Individualized Developmental Care and Assessment Program (NIDCAP), which is based on the concept of newborn or infant competence and focuses on respecting the individuality of the very tiny human being and his or her family. The NIDCAP incorporates some principles of developmental care, such as positioning and the use of incubator covers to shield infants. Additionally, the hospital encourages breastfeeding and parents’ involvement during a child’s daily care and provides support to the families during the NICU stay through a multidisciplinary team, which consists of neonatologists, nurses, psychologists, social workers, and physiotherapists. The usual physiotherapy care offered during the NICU stay includes at least one visit to guide parents during the daily activities in the neonatal unit. Before hospital discharge, parents are also invited to a session called “*going home*”, where a nurse and a physiotherapist give specific reminders of the basics of infant care at home (breastmilk storage and formula feeding, kangaroo care at home, baby massage, and guidance about positioning, feeding, and sleeping at home). After hospital discharge, if the infant is considered high-risk, sporadic sessions with a PT for individual assistance are arranged. In those cases, the number and type of activities were recorded.

**Early physiotherapy intervention program**. In addition to standard care, the intervention group received the early physiotherapy intervention program during the NICU stay and after hospital discharge. The intervention is based on traditional developmental care programs to improve preterm infants’ development outcomes [22,39], and it was designed with a preventative approach. It aimed to improve preterm infants’ motor development, enhance the parent–infant relationship, and teach parents about preterm infants’ cues and management strategies (see Figure 1).

Intervention at the NICU: The intervention commenced after the infant’s 32 weeks PMA and before term-equivalent age, during the NICU stay, once the preterm infant was medically stable. In the case of infants born after 32 weeks PMA, the intervention started 4–5 days after birth. 

Firstly, parents received a total of six education sessions for 2–3 weeks. Each session lasted 1 h, and approximately two sessions were held each week. The education sessions consisted of teaching parents to understand their infants’ communication cues, recognize signs of distress, and how to respond sensitively to them. The sessions were divided into different topics. In the first session, the therapist explained the characteristics and necessities of the preterm infant regarding light, noise, positioning (in prone, supine, and lateral), and touch, and she also described infants’ cues and distress signs and how to respond sensitively to them. In the second session, the therapist taught parents how to perform the tactile and kinesthetic stimulation and involved them actively to perform it themselves. The subsequent two sessions focused on understanding and identifying infants’ cues during daily activities, such as bathing, diaper changing, and feeding, and learning how to interact sensitively with the infant and incorporate some recommendations during these activities to enhance infants’ development. In the last two sessions, the PT explained the typical development of the preterm infant, the recommended toys to enrich their development, and different positions to stimulate and play with the baby, promoting antigravity skills and facilitating hands to the midline and toy manipulation. Although the program can involve the whole family, the mother and the father were the principal recipients of the intervention.

At the same time, during the second education session, parents were asked to carry out the tactile and kinesthetic stimulation. The protocol was inspired by Fucile and Gisel’s (2010) tactile/kinesthetic sensorimotor stimulation model. Each stimulation consisted of 10 min of slow tactile stimulation of the baby, applying moderate pressure stroking with both hands. During the tactile stimulation, the baby was placed in the prone position. After that, 5 min of kinesthetic stimulation was provided, performing passive movements in the infant’s hip, knee, ankle, shoulder, elbow, and wrist. The infant was placed in a supine position for this stimulation, either in the crib or on the parent’s lap. Parents were encouraged to provide preterm baby stimulation sessions twice a day, at least 30 min after feeding and 2 h after the previous one, for ten days in a 15-days period. If the infant showed any sign of stress or an adverse behavior (e.g., an increase in tone, hiccupping, fluctuating behavior state), parents were asked to stop the stimulation to calm the infant, or the session was terminated. During the NICU intervention, parental compliance was registered by the physiotherapist.

Intervention at home: After hospital discharge, the program continued at the family home, from term-equivalent age until 2 months corrected age. During this period, the physiotherapist presented to parents a program of activities for the child, involving them actively in the intervention. The activity program was designed to be included in the family’s routine by providing opportunities to experience different positions and movements appropriate for the infants’ development. The PT proposed varying degrees of activities (by focusing on toys or the parents’ face) in different positions to encourage the infants’ movements to bring the head and extremities toward the midline, improve head and postural control, promote antigravity skills, and facilitate toy manipulation. A maximum of four activities was given at each appointment. Activities such as holding the infant while gently encouraging them to bring his or her hands to their midline, prone play (tummy time), and placing small toys on infants’ hands to promote object exploration were included. The PT provided a demonstration to explain to parents how to complete each activity with their infant. Parents were encouraged to carry out the activities for 15–20 min, 2 times per day (with a separation of 4–5 h between), 5 days per week. With the home program, a daily diary was given to parents to document if they accomplished each activity and the time performed every day. The families received one or two visits per month by the PT (in total, a maximum of four visits at home) to progress with the program, answer questions, and explain the importance of promoting infants’ development. Each home visit lasted approximately 1 h. Additionally, during the home program, parents received a short telephone message weekly to encourage them to continue completing the activities. Parents were also asked to keep the activities diary during the intervention at home. 

Participants received written materials with pictures summarizing each educational session and describing the stimulation procedure and the home activity program to develop them with their infant at home and increase their accomplishment and adherence to the program. 

### 2.6. Measures

After enrolment, perinatal, demographic, and clinical data were collected through infants’ medical notes and parental interviews. Recorded data included the following: gender, birth weight, head circumference at birth, gestational age, twin birth, small for gestational age (SGA), mode of delivery, assisted reproduction, mother age and educational attainment, father age and educational attainment, length of hospitalization, weight and head circumference at hospital discharge, and medical interventions during the neonatal period (oxygen and antibiotic use). 

The **primary outcome measure** consisted of the Spanish version of the Alberta Infant Motor Scale [36]. It is a standardized and discriminative scale for gross motor development and assesses infants’ motor abilities, quality of posture, and movement in four positions: prone, supine, sitting, and standing. The four positional scores are calculated to determine infants’ total AIMS score [36,40]. The evaluations were performed at 2- and 8-months corrected age. Assessments were scheduled at the hospital and performed by four independent and experienced PTs. The assessors were blinded regarding group assignments. All examinations were video-recorded and regularly reviewed to maintain consistent scoring. Additionally, to ensure that the four examiners’ administration and scoring of the AIMS were consistent, before commencing the study they practiced assessing some preterm infants independently at 2 and 8 corrected months. If there were disagreements, they reviewed the AIMS manual to resolve the doubts and reach an agreement.

The **secondary outcome measures** consisted of global development and the parental stress index. The first one was the Spanish version of the Ages and Stages Questionnaires Third Edition (ASQ-3) [41,42]. It is a child developmental progress screening questionnaire based on milestones that should be achieved between 0- and 66-months old. The ASQ-3 was designed to monitor a child’s development in five domains: gross motor, fine motor, problem-solving, communication, and personal-social. Each area has six items; each scored on a 3-point scale of 10, 5, or 0. The scores are summed to give scores for each subscale between 0 and 60 points. A higher score indicates better development [43,44,45]. The assessments were made at the infant’s 1- and 8-months corrected age. The 2 months age-specific sheet for children between 1 month 0 days to 2 months 30 days was used at 1 month corrected age. The 8 months age-specific sheet for children between 7 months 0 days to 8 months 30 days was used at 8 months corrected age. The questionnaires were completed by parents through a telephone call from a blinded researcher. Research shows that the ASQ has been successfully used for follow-up and assessment of premature and at-risk infants and children [46,47] and also that parents accurately report skills their children can perform and rarely misrepresent their children’s developmental acumen [44].

Furthermore, the Spanish version of the Parenting Stress Index-Short Form (PSI-SF) was used to assess the parental stress index [48,49]. This tool measures the feelings of stress a person experiences regarding his or her role as a parent. It is a self-report measure comprising 36 items in its short version to which parents must respond on a 5-point Likert-type scale. The PSI-SF consists of three subscales of 12 items each: the *Parental Distress* subscale determines distress experienced by parents in exercising the parental role; the *Parent–Child Dysfunctional Interaction* subscale focuses on perceptions that parents have as to what extent their child meets expectations or not, and the degree of reinforcement their child provides them; and, the *Difficult Child* subscale assesses how parents perceive the ease or difficulty of controlling their children in terms of their behavioral traits. From the sum of these three subscales, a final overall score is obtained called *Total Stress* [48,49,50]. The assessments were made at the infant’s 3 months corrected age, and aimed to assess the parents’ post-intervention stress index. The questionnaire was completed by parents through a telephone call from a blinded researcher. The researcher asked them to write down the 5 answer options to facilitate the task of answering to mothers. 

**Parental compliance** was defined as the number of activities performed by parents and the time spent on the home program. The parents were asked to document in a daily diary the time and activities accomplished in each phase of the home program. When assessing the parents’ compliance, executed activities, minutes, and days of the home program were calculated from the daily diaries and categorized as “good compliance” or “poor compliance”. Compliance was rated good when parents provided at least the recommended dose (5 days per week, at least 2 activities of the program), and poor if they did not accomplish the proposed dose. 

At 8 months corrected age, parents were asked about additional physiotherapy interventions received between 2- and 8-months corrected age. This information was used to analyze whether there were differences between groups regarding additional physiotherapy interventions received during this period. 

### 2.7. Statistical Analysis

All data were analyzed using the Statistical Package for the Social Science (SPSS Version 26; SPSS, Inc, Chicago, IL, USA). Categorical variables were compared by Fisher’s exact test and chi-squared test. Shapiro–Wilk’s test verified the normality of distribution. The Student’s *t*-test for independent samples was used for normally distributed data and the non-parametric Mann–Whitney test for non-normal distributed data. Bonferroni correction was performed for every hypothesis related to the primary outcome measure. A level of significance of 0.05 was used for all the statistical tests. 

Baseline statistics for perinatal, demographic, and clinical characteristics were performed to test a priori baseline differences between the two groups. After that, to verify the effect of the early physiotherapy intervention program versus standard care, differences between groups were calculated in primary (AIMS total and subdomains) and secondary (ASQ-3 domains and PSI-SF) outcome measures. Additionally, the effect size was computed using Cohen’s d or the r test, depending on whether the data showed normal or non-normal distribution, respectively [51,52,53]. The between-groups effect size was calculated for AIMS, ASQ-3, and PSI-SF. Commonly used criteria specify that a Cohen’s d value below 0.2 is regarded as no effect, a value of 0.2 as a small effect, a value of 0.5 as a medium-sized, a value above 0.8 as a large effect, and above 1.3 as a very large effect [51]. The effect size for the r test was calculated using the following formula: $r=\frac{Z}{\sqrt{N}}$, where *N* is the total of the samples. The values of *r* were considered small when they were higher than 0.1, medium when higher than 0.3, and large when higher than 0.5 [53].

Additionally, The ASQ-3 scores were categorized for each domain as “high-risk development”, “need for follow-up”, and “correct development”, following the standard criteria. These results were analyzed through the chi-squared linear trend.

Through longitudinal analyses, changes in primary and secondary outcome measures were calculated between the first and the second assessments. Multivariate tests were used. 

Finally, infants’ primary and secondary outcome measures were analyzed regarding parental compliance within the intervention group. The parental stress index was also analyzed regarding their compliance. After dichotomizing parental compliance as “high compliance” and “poor compliance”, differences between groups were calculated, as well as the effect size. 

## 3. Results

Initially, 69 preterm infants were assessed for eligibility. A total of 21 children were then excluded because they did not meet the inclusion criteria (n = 15), or parents declined to participate in the study (n = 6). The remaining 48 children were recruited and randomized into intervention (n = 24) and control group (n = 24). The recruitment was considered to have ended when all the sample size was gathered. 

One infant in the control group passed away after hospital discharge, leaving 23 infants to complete the first assessment at 1 month corrected age. Furthermore, 1 infant from the control group was not available for the evaluation at 2 months corrected age, resulting in 22 infants with the post-intervention assessment. During the follow-up, 1 infant was lost due to parents’ lack of availability, leaving 21 infants in the control group for the mid-term analyses (8 months corrected age). In the intervention group, three children withdrew before completing the intervention. After performing all the assessments, one infant in the intervention group was excluded from all analyses (genetic mutation diagnosed). A total of 20 infants in the intervention group were analyzed at all the assessment measures (see Figure 2). 

The baseline characteristics of infants and their parents are reported in Table 1 and Table 2. No statistically significant differences were found between the two groups at baseline (*p*-value > 0.05). 

The early physiotherapy intervention at the neonatal unit commenced and finished at a mean age of 35- and 36-weeks PMA, respectively. At the beginning of the intervention, the minimum age was 32 weeks PMA, and the maximum age at the end of the intervention was 39 weeks PMA. Parents of infants allocated to the intervention group received a mean of five education sessions, each of 45 min approximately. All the infants received the recommended dose of tactile and kinesthetic stimulation, starting after the second parent education session. Infants received a mean of 17.94 ± 4.05 min of stimulation per day, in one or two sessions, during at least 10 days in a 15-day period. No parent reported stress signs or adverse behaviors in their infants during the stimulation. 

After hospital discharge, at 40 weeks PMA, all the infants allocated in the intervention group started the home program. They received a mean of three sessions with the PT, each of 45 min approximately. The program lasted 2 months in all cases. The first phase commenced at 40 weeks PMA until 1 month corrected age, and the second phase started at 1 month corrected age until 2 months corrected age. Although both parents were invited to perform the intervention, the mothers were the ones who performed almost all the interventions. 

### 3.1. Primary Outcome Measure (AIMS)

Infants’ gross motor development was assessed after the intervention period, at 2 months corrected age, and at 8 months corrected age. As shown in Table 3, there were no statistically significant differences between both groups in mean AIMS total or four positional scores in either of the assessments nor within the two groups from the first to the second assessments. The effect size in AIMS total score at 2 months corrected age was small (r: −0.277).

The longitudinal analysis did not show statistically significant differences between the two groups from 2 months to 8 months corrected age in any AIMS positional scores or the total score. 

### 3.2. Secondary Outcome Measures

**ASQ-3.** After the first phase of the home program was completed, at 1 month corrected age, infants’ global development was assessed for the first time. When comparing the mean differences in all subscales of the ASQ-3, fine motor, problem-solving, communication, and personal-social domains showed statistically significant differences between groups (*p*-value < 0.05) (see Table 3). Gross motor development did not differ statistically between groups. The differences in fine motor (r: −0.641), problem-solving (r: −0.416), and communication (r: −0.418) development showed a large effect size between groups at 1 month corrected age (see Table 3). 

At the second assessment, at infants’ 8 months corrected age, there were no statistically significant differences between both groups in the five domains of the ASQ-3, and neither were in the categorizations of global development (see Table 3). 

Global development was categorized as high-risk of development delay, the need for follow-up, or correct development. As shown in Table 4, there were statistically significant differences between groups in fine motor, problem-solving, and communication domain categorizations. 

**PSI-SF.** The parental post-intervention stress index was assessed at 3 months corrected age. Mothers were the principal recipients, so 19 mothers in the control group and 16 in the intervention group were assessed with the PSI-SF. As shown in Table 5, there were no statistically significant differences between both groups in mean SPI-SF total or the three subdomain scores.

**Parental compliance.** The daily diary of the home program was completed by 12 (75%) mothers in the intervention group. Four mothers (25%) did not complete the diary correctly, so they were excluded from the analyses. Five (41.67%) mothers demonstrated good compliance, and seven (58.33%) had poor compliance. The results found an association between parents’ compliance and infants’ development. Infants of mothers with good compliance showed higher scores on the first assessments of AIMS or ASQ-3 compared with those with poor compliance. However, there were no statistically significant differences. The between-groups effect size was medium for total AIMS score at 2 months corrected age (r: 0.43; *p*-value 0.121), for prone (r: 0.39; *p*-value 0.152) and supine (r: 0.45; *p*-value 0.232) positions. Communication development at 1 month corrected age by ASQ-3 showed a large size effect (r: 0.64; *p*-value 0.014) and personal-social development displayed a medium size effect (r: 0.42; *p*-value 0.121). Gross and fine motor development by ASQ-3 showed a medium size effect (r: 0.41, *p*-value 0.152 and r: 0.30, *p*-value 0.336). 

Mothers with good compliance had lower scores in all domains of the PSI-SF. Parental distress score was statistically lower in mothers who showed good compliance than mothers with poor compliance (*p*-value 0.013), with a very large size effect (Cohen’s d: 1.76). The total stress score was not statistically different depending on parental compliance (*p*-value 0.073), but it showed a very large size effect (Cohen’s d: 4.39). 

Regarding additional physiotherapy interventions received between 2- and 8- months corrected age, ten infants (50%) in the intervention group and eight (34.8%) in the control group were given other interventions. There were no statistically significant differences between groups in the proportion of additional physiotherapy interventions received. No more information was asked about the kind of intervention received or the dose.

## 4. Discussion

Most of the early interventions provided to preterm infants aim to prevent infants’ neurodevelopment delays in the short- and long-term. In this framework, the main goal of the early physiotherapy intervention was to prevent preterm infants’ motor and global developmental delay and decrease the parental stress index. Results achieved in post-intervention and short-term assessments suggest that the intervention provided, from the neonatal period until 2 months corrected age, may not affect the overall gross motor development of infants born preterm or the parental stress index. Nevertheless, the intervention might improve infants’ fine motor, communication, problem-solving, and personal-social development at post-intervention follow-up. Furthermore, although mothers showed poor compliance with the home program, the results might suggest that preterm infants’ development at 1- and 2- months corrected age and mothers’ stress index could be related to good parental compliance. 

A recent systematic review [18], evaluating neonatal therapies for preterm infants, found that the daily parent-delivered motor intervention improves infants’ motor and cognitive outcomes in the short- and possibly in long term. This approach includes teaching a parent to provide postural support and opportunities for movement with assistance during a parent–infant interaction. Likewise, interventions starting during the NICU period and continuing beyond the neonatal period have the strongest effect on infants’ long-term motor development [17,21,22]. Although the early physiotherapy intervention described in the present study follows the recommendation by Khurana et al., (2020), our findings were not consistent with a more optimal gross motor development at 2- and 8-months corrected age. A recently published randomized controlled trial concluded that parent-administered individualized early motor intervention programs in the NICU could substantially affect motor development in infants born preterm if the intervention dosage is at least as high as 222 min [54]. Another piece of research published in 2001 analyzed the effectiveness of a physiotherapy program on at-risk preterm infants. The program was performed from 40 weeks PMA until 4 months corrected age, and the results showed more significant improvements in infants’ motor development after receiving the physiotherapy program [32]. Therefore, the short dose or intensity of the early physiotherapy intervention described in the present study might be one of the reasons for the lack of group differences in preterm infants’ motor development [22,29,32,55]. It is well known that early intervention results in brain structure reorganization, and hence, improved outcomes [17], and it has also been well considered that increased intervention dosage might be attributed to those changes in the central nervous system [54,56,57].

As the present intervention targeted preterm infants and their parents, those played a significant role in carrying out the program. In fact, the primary caregivers, in all cases the mothers, were actively involved in the intervention: they received information regarding their infants’ development and the aims for each activity; and were also taught to enhance their infants’ development. Therefore, the program’s effectiveness relied partially on the mothers’ understanding and compliance [22,31,58]. The present study followed strategies known to be helpful to increase parents’ compliance and adherence, such as including the exercises into daily routines, providing written instructions, and demonstrating the activities with the child [59,60,61,62]. Nevertheless, mothers showed poor compliance with the home program. Similarly, a recently published study assessing the feasibility of a novel physiotherapy intervention for preterm infants incorporating participation goals and telehealth delivery showed that parental adherence to dose was lower than expected as infants attended an average of half of the prescribed session dose. The authors reported that paternal mental health was the most common reason for non-attendance [63]. Another research suggested that parental adherence depends on factors such as self-efficacy, perception of barriers and ability to perform the program [59]. Those aspects were not considered during the development of the present project, and it might have influenced the results related to parental compliance and, consequently, the effectiveness of the early physiotherapy intervention. Therefore, future research should consider these findings and study the importance of assisting parents in achieving good compliance during the home programs.

The present study showed that mothers with good compliance with the home program had lower stress levels in the parental distress area, which is determined by the experience of the parental role [49,50,64]. Although parenting stress is known to be closely correlated with children’s behavioral development [65], the evidence suggests that parents’ involvement in the early intervention reinforces parents’ role and decreases their stress index [10,15,66,67]. A recently published qualitative study concluded that parents’ involvement in infants’ care is associated with increased empowerment. Those parents also develop more coping strategies to deal with prematurity and challenges in the parenting role [16]. Similarly, a randomized controlled trial studied whether early intervention focused on sensitizing affected the parenting stress among mothers and fathers of preterm infants. The authors found that this kind of intervention reduces maternal stress and positively influences mothers’ perceptions of their children’s adaptability and happiness [65]. On the contrary, it might also be suggested that less stressed mothers were more able to comply with the proposed program, and thus produced better outcomes for the infants’ development and mothers’ stress index. However, more research is needed in this aspect to study these hypotheses. 

Regarding the infants’ motor and global development, there is abundant evidence supporting that certain brain circuits’ structures can change in response to environmental stimuli [68,69]. In fact, the brain’s structural-developmental processes result from a continuous interaction between experience, activity, environment, and genes [2]. For instance, postural control demands, which are usually presented to infants by their caregivers, are also crucial when structuring the central nervous system [3]. The present study showed higher percentages of the correct development of communication, problem-solving, personal-social, and fine motor skills in the intervention group compared to the control group. Furthermore, infants of parents that showed good compliance to the early physiotherapy intervention displayed more optimal gross motor and global development. Therefore, our results might suggest that infants who received the recommended dose of the home program could show more positive gross motor and global development. However, these results should be interpreted prudently as no statistical adjustments were performed for the secondary outcomes.

A strength of this research is that the early physiotherapy intervention was designed according to the latest recommendations: it started during the highly neuroplastic period in the NICU and continued in the infant’s first months of life, and it encouraged early parental engagement and focused on providing movement opportunities to the infant born preterm. Moreover, all the included infants were born preterm and did not have any brain injury during the neonatal period or later, which reduced the heterogeneity of the participants. However, several limitations should be considered. Firstly, the possible interpretations might be hard to justify due to the lack of pre-intervention assessments related to infants’ gross motor or global development and parental stress index. While it is true that there are no data to ensure the probable effect of the early physiotherapy intervention program, it is reasonable to say that infants started from a similar situation because all the assessed parameters at baseline—factors that could influence infants’ neurodevelopment [70]—showed there were no differences between groups. For that reason, the results suggest that displayed differences could be a consequence of the intervention. Secondly, in the present study, researchers could not control the criteria for deciding whether or not a child received additional therapy between 2- and 8-months corrected age. Consequently, infants could have received additional interventions based on different criteria, which could have biased these results. Furthermore, the hospital’s protocols and logistics were unavoidable barriers when we proposed administering a different assessment tool and follow-up time points. Hence, it was impossible to use other assessment tools that are more sensitive for the preterm infants population, such as the TIMP. Another weakness was that relevant adjustments were only carried out with the primary outcome (the AIMS), and the results related to the secondary outcomes must be interpreted prudently. Multiplicity refers to the potential inflation of the type I error rate as a result of multiple testing. So, further confirmatory studies are needed to support the findings on the infants’ global development and parents’ stress index. Moreover, a multiple linear regression analysis between sociodemographic variables, parental involvement in the intervention group, and infants’ gross motor development in the post-intervention assessment could not be performed due to the limited sample size. Future research should include a larger sample size to be able to build more powerful statistic models. Regarding parental compliance, it was assessed only at the end of the home program by a non-standardized tool. Thus, more research is worthwhile looking at the parent diaries as the home program intervention is provided to encourage higher parental compliance, if necessary, for example, by increasing the frequency of visits. Moreover, further investigation is needed to establish standardized and validated parental compliance recording tools. Finally, the present study included infants of parents that were in the NICU for 6 or more hours a day. This must be considered when interpreting the results. While it is essential to engage parents in early intervention, it is also important to consider that those who are unable to be present might be the ones that need to be targeted because they might benefit from the intervention and the education demonstrating better intervention effects. Therefore, future research should take this consideration into account. 

## 5. Conclusions

The results of this RCT suggest that the early physiotherapy intervention was not effective on preterm infants’ motor or global outcomes nor on parental stress index. These results might be related to the dose or intensity of the intervention, but also to poor parental compliance. Furthermore, there is the possibility that parental higher compliance impacted on mother’s stress index positively, as mothers with high compliance showed a lower stress index. Further research should consider these findings. 

## Figures and Tables

**Figure 1 children-09-00895-f001:**
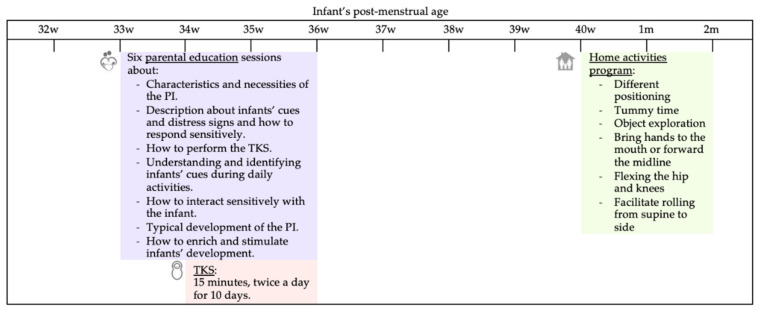
Example of the intervention protocol for a preterm infant born at 31 weeks of gestational age. The protocol was adapted depending on the infant’s gestational weeks at birth. PI: preterm infant; TKS: tactile-kinesthetic stimulation.

**Figure 2 children-09-00895-f002:**
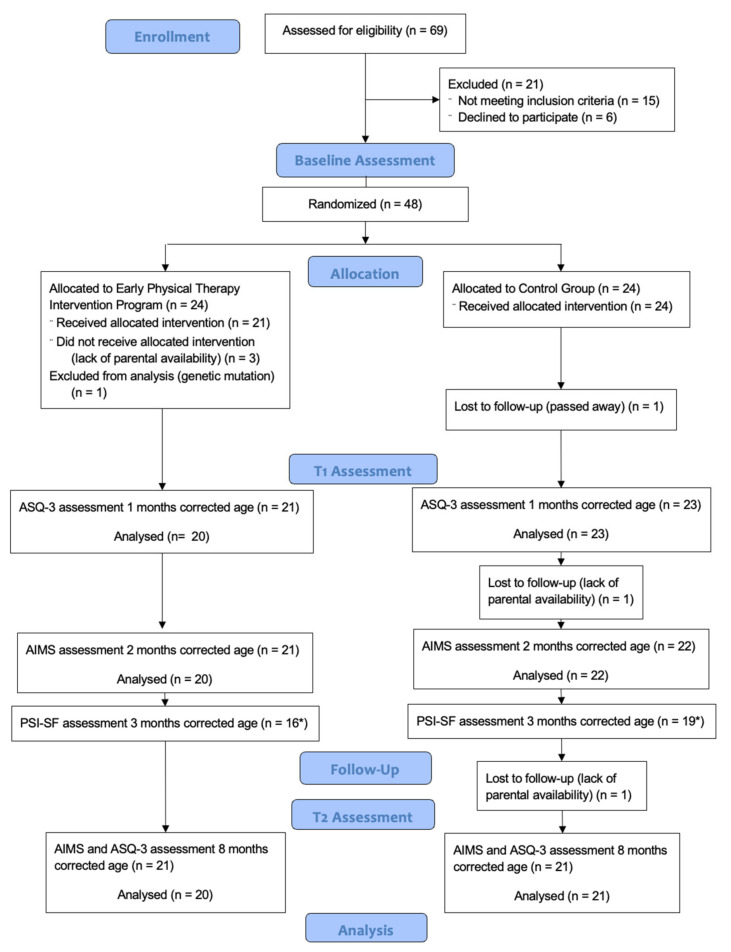
Flow-chart. * Mothers.

**Table 1 children-09-00895-t001:** Sociodemographic and baseline characteristics of infants in the control group (CG), and intervention group (IG).

Children Characteristics	IG (n = 24)	CG (n = 24)	*p*-Value
**Gestational age at birth, M (SD) (weeks)**	31.84 (1.82)	32.05 (1.59)	0.675 ^a^
**Prematurity classification, n (%)**			
Moderate premature 32 < 34	13 (48.1)	14 (51.9)	0.771 ^c^
Very premature 28 < 32	11 (52.4)	10 (47.6)	
**Intrauterine growth restriction, n (%)**	4 (16.7)	6 (25.0)	0.724 ^b^
**Birth weight, M (DE) (g)**	1462.46 (437.27)	1590.79 (331.21)	0.258 ^a^
**Cranial circumference at birth, M (SD) (cm)**	27.75 (2.61)	28.48 (1.51)	0.247 ^a^
**Gender, n (%)**			
Girls	12 (50.0)	8 (33.3)	0.242 ^c^
Boys	12 (50.0)	16 (66.7)	
**Twins, n (%)**	8 (33.3)	8 (33.3)	1.000 ^c^
**Cesarean birth, n (%)**	14 (58.3)	12 (50.0)	0.562 ^c^
**Assisted reproduction, n (%)**	9 (37.5)	5 (20.8)	0.204 ^c^
**Antibiotic therapy, n (%)**			
Yes	10 (41.7)	8 (33.3)	0.551 ^c^
No	14 (58.3)	16 (66.7)	
**Oxygen therapy, n (%)**			
Yes	4 (16.7)	4 (16.7)	1.000 ^b^
No	20 (83.3)	20 (83.3)	
**Jaundice with phototherapy, n (%)**			
Yes	17 (70.8)	17 (70.8)	1.000 ^c^
No	7 (29.2)	7 (29.2)	
**Caffeine, n (%)**			
Yes	13 (54.2)	14 (58.3)	0.771 ^c^
No	11 (45.8)	10 (41.7)	
**Hospital admission days, M (SD)**	37.00 (19.12)	32.25 (17.04)	0.368 ^a^
**Weight at hospital discharge, M (SD) (g)**	2086.67 (283.05)	2096.46 (236.83)	0.897 ^a^
**Cranial circumference at discharge, M (SD) (cm)**	32.42 (3.91)	31.38 (1.24)	0.224 ^a^

^a^ Student’s *t*-test. ^b^ Fisher’s exact test. ^c^ Pearson’s chi-squared test.

**Table 2 children-09-00895-t002:** Sociodemographic and baseline characteristics of the mothers and fathers of the intervention group (IG) and control group (CG).

Mother Characteristics	IG (n = 20)	CG (n = 20)	*p*-Value
**Age, M (DE) (years)**	33.80 (5.95)	32.20 (6.26)	0.413 ^a^
**Education level, n (%)**			
Primary	1 (5.0)	2 (10.0)	0.160 ^b^
High school	2 (10.0)	7 (35.0)	
Medium	11 (55.0)	6 (30.0)	
Undergraduate	6 (30.0)	5 (25.0)	
**Father characteristics**	**IG (n = 20)**	**CG (n = 19)**	** *p* ** **value**
**Age, M (DE) (years)**	33.40 (4.83)	33.74 (7.24)	0.866 ^a^
**Education level, n (%)**			
Primary	0 (0.0)	1 (5.3)	0.259 ^b^
High school	3 (15.0)	3 (15.8)	
Medium	11 (55.0)	12 (63.2)	
Undergraduate	6 (30.0)	3 (15.8)	

^a^ Student’s *t*-test. ^b^ Linear chi-squared test.

**Table 3 children-09-00895-t003:** Alberta Infant Motor Scale (AIMS) at 2- and 8-months corrected age, and Ages and Stages Questionnaire—Version 3 (ASQ-3) at 1- and 8-months of corrected age in the intervention group (IG) and control group (CG).

AIMS	2 Months Corrected Age					8 Months Corrected Age				
	IG (n = 20)	CG (n = 22)	*p*-Value ^a^	Z Mann Whitney	Effect Size ^b^	IG (n = 20)	CG (n = 21)	*p*-Value ^a^	Z Mann Whitney	Effect Size ^b^
Prone, M (SD)	2.30 (1.13)	2.73 (0.99)	0.121	−1.550	−0.239	12.30 (4.04)	12.33 (3.53)	0.875	−0.157	−0.025
Supine, M (SD)	3.15 (0.37)	3.36 (0.58)	0.190	−1.310	−0.202	7.30 (1.42)	7.29 (1.42)	0.989	−0.013	−0.002
Sitting, M (SD)	1.05 (0.22)	1.05 (0.38)	0.980	−0.025	−0.004	7.55 (2.95)	8.76 (2.05)	0.230	−1.201	−0.188
Standing positions, M (SD)	1.25 (0.64)	1.55 (0.60)	0.118	−1.563	−0.241	3.10 (0.97)	3.67 (1.02)	0.063	−2.921	−0.456
Total score, M (SD)	7.80 (1.20)	8.55 (1.50)	0.073	−1.793	−0.277	30.25 (7.15)	32.10 (6.83)	0.396	−1.045	−0.163
**ASQ-3**	**1 Month Corrected Age**					**8 Months Corrected Age**				
	**IG (n = 20)**	**CG (n = 23)**	** *p* ** **Value ^a^**	**Z Mann Whitney**	**Effect Size ^b^**	**IG (n = 20)**	**CG (n = 23)**	** *p* ** **Value ^a^**	**Z Mann Whitney**	**Effect Size ^b^**
Gross Motor, M (SD)	50.00 (5.85)	49.13 (7.18)	0.830	−0.215	−0.033	38.25 (14.63)	41.09 (13.73)	0.532	−0.625	−0.095
Fine Motor, M (SD)	53.50 (5.16)	42.17 (8.64)	0.000	−4.202	−0.641	55.50 (6.05)	54.78 (5.93)	0.517	−0.648	−0.099
Problem Solving, M (SD)	49.25 (7.83)	37.39 (14.91)	0.006	−2.725	−0.416	52.50 (7.69)	55.00 (5.22)	0.351	−0.932	−0.142
Personal-Social, M (SD)	46.25 (5.10)	41.96 (6.70)	0.049	−1.972	−0.301	47.00 (11.17)	50.87 (8.61)	0.263	−1.120	−0.171
Communication, M (SD)	38.00 (12.61)	26.52 (13.52)	0.006	−2.740	−0.418	51.00 (9.40)	51.30 (7.10)	0.870	−0.163	0.025

^a^ Mann-Whitney U test. ^b^ Mann Whitney size effect: <0.1 negligible; 0.1–0.3 small effect; 0.3–0.5 medium; >0.5 large effect.

**Table 4 children-09-00895-t004:** Results of Ages and Stages Questionnaire—version 3 (ASQ-3) at 1 month corrected age.

ASQ-3 at 1 Month Corrected Age	IG (n = 20)	CG (n = 23)	*p*-Value ^a^
**Gross Motor, n (%)**			0.588
High-risk	2 (10.0)	4 (17.4)	
Follow-up	4 (20.0)	4 (17.4)	
Correct	14 (70.0)	15 (65.2)	
**Fine Motor, n (%)**			0.003
High-risk	0 (0)	4 (17.4)	
Follow-up	1 (5.0)	7 (30.4)	
Correct	19 (95.0)	12 (52.2)	
**Problem Solving, n (%)**			0.004
High-risk	0 (0)	6 (26.1)	
Follow-up	1 (5.0)	4 (17.4)	
Correct	19 (95.0)	13 (56.5)	
**Personal-Social, n (%)**			0.079
High-risk	0 (0)	2 (8.7)	
Follow-up	5 (25.0)	9 (39.1)	
Correct	15 (75.0)	12 (52.2)	
**Communication, n (%)**			0.003
High-risk	2 (10.0)	9 (39.1)	
Follow-up	6 (30.0)	10 (43.5)	
Correct	12 (60.0)	4 (17.4)	

^a^ Linear chi-squared test.

**Table 5 children-09-00895-t005:** Results of parenting Stress Index (PSI-SF) post-intervention in the intervention group (IG), and control group (CG).

Parenting Stress Index (PSI-SF)	IG (n = 16)	CG (n = 19)	*p*-Value ^a^	Size Effect ^b^
Parental Distress, M (SD)	26.38 (10.51)	27.05 (8.46)	0.834	0.07
Parent–Child Dysfunctional Interaction, M (SD)	19.69 (5.63)	18.84 (4.63)	0.629	0.16
Difficult Child, M (SD)	21.75 (7.79)	21.21 (5.35)	0.810	0.08
Total Stress, M (SD)	68.31 (21.94)	67.11 (16.24)	0.853	0.06

^a^ Student’s *t*-test. ^b^ Cohen’s d: 0.20: small effect; 0.50: medium size; 0.8: large effect; >1.3: very large.

## Data Availability

Not applicable.

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
