# Peer review of "Early Physiotherapy Intervention Program for Preterm Infants and Parents: A Randomized, Single-Blind Clinical Trial"

_children, 2022, doi:10.3390/children9060895_

Round 1
Reviewer 1 Report
General comments: This is a well-designed and well-written study on an important topic. A clear rationale was provided, the study was adequately powered, outcome assessments were blinded, and there was no contamination of the control group. The program provided followed general principles that have been reported in the literature, and outcome measures were well chosen. Statistical analyses were appropriate and results were interpreted accurately. The lack of group differences was mainly attributed to poor compliance and the short length of the intervention; however, the authors missed an opportunity to reflect more on the intervention elements and their intensity and how these may have influenced the limited results. If early intervention is deemed to be so important for early brain development and we fail to find effects, we need to be very critical of the elements of our program as well as the dosage.
One of the major problems, perhaps underrecognized in preterm infants, is the lack of muscle development and strength, in part due to the fact that they did not experience the “resistance exercises” babies experience in late pregnancy when pushing against the walls of the uterus. While tactile input is critical early on, activities to encourage infant movement against gravity should be started as soon as it is safe to do so (even in the NICU). Passive exercises of the limbs are not helpful. Handling and positioning were discussed, as one of multiple activities – and I assumed these were to promote antigravity skills as well as facilitating hands to mouth and midline and toy manipulation. Perhaps this should have a stronger focus in this group of children. The other issue that needs to be raised is feasibility - if compliance was poor, this should be discussed because these were likely motivated parents and the compliance was surprisingly poor. Only 2 activities is not very demanding, and still it was too much for some. Associations are not directional, but it is possible that less stressed parents were able to comply more and thus produced better outcomes – rather or in addition to the other way around? It would be interesting to evaluate which activities parents tended most to do with the babies to see if that influenced the developmental domains that changed.
It is important to publish high quality studies such as this one that present both positive and negative results because all contribute to our evidence base and challenge us to find more optimal strategies to promote development and support families.
Author Response
Reviewer 1
Comments and Suggestions for Authors
General comments: This is a well-designed and well-written study on an important topic. A clear rationale was provided, the study was adequately powered, outcome assessments were blinded, and there was no contamination of the control group. The program provided followed general principles that have been reported in the literature, and outcome measures were well chosen. Statistical analyses were appropriate and results were interpreted accurately. The lack of group differences was mainly attributed to poor compliance and the short length of the intervention; however, the authors missed an opportunity to reflect more on the intervention elements and their intensity and how these may have influenced the limited results. If early intervention is deemed to be so important for early brain development and we fail to find effects, we need to be very critical of the elements of our program as well as the dosage.
Authors: Thank you very much for your kind comments. We have considered your suggestion and added some lines at the end of the second paragraph of the discussion to emphasize these comments (lines 516-530). Changes are highlighted inside the manuscript.
One of the major problems, perhaps underrecognized in preterm infants, is the lack of muscle development and strength, in part due to the fact that they did not experience the “resistance exercises” babies experience in late pregnancy when pushing against the walls of the uterus. While tactile input is critical early on, activities to encourage infant movement against gravity should be started as soon as it is safe to do so (even in the NICU). Passive exercises of the limbs are not helpful. Handling and positioning were discussed, as one of multiple activities – and I assumed these were to promote antigravity skills as well as facilitating hands to mouth and midline and toy manipulation. Perhaps this should have a stronger focus in this group of children.
Authors: We are grateful for your comments. We have specified that positioning during NICU stay was promoted in supine, prone and lateral; and that home activities promoted antigravity skills and toy manipulation, among others. Changes are highlighted inside the manuscript (lines 205-206; 215 and 243).
The other issue that needs to be raised is feasibility - if compliance was poor, this should be discussed because these were likely motivated parents and the compliance was surprisingly poor. Only 2 activities is not very demanding, and still it was too much for some. Associations are not directional, but it is possible that less stressed parents were able to comply more and thus produced better outcomes – rather or in addition to the other way around? It would be interesting to evaluate which activities parents tended most to do with the babies to see if that influenced the developmental domains that changed.
Authors: Thank you so much for sharing your point of view. We agree with you and add some lines in the discussion to develop this argument. Changes are highlighted inside the manuscript (lines 539-551)
We did evaluate the activities that parents tended most to do, but there were no correlations with the developmental domains changed in infants.
It is important to publish high quality studies such as this one that present both positive and negative results because all contribute to our evidence base and challenge us to find more optimal strategies to promote development and support families.
Authors: Thank you so much for your consideration.

Reviewer 2 Report
See attached file

Author Response
Reviewer 2
Outcomes:
While the rationale for providing an intervention in NICU and home is clear it is less clear as to the age choices for outcome measurement.
For example, there is no pre intervention assessment of infants. Intervention finishes at 2 months corrected age (the home component) – the AIMS is performed at 2 months, i.e post intervention but the ASQ is performed at 1 month. It is unclear as to why one of the outcome measures would be completed before completion of the actual intervention.
The 8 month outcome assessments are appropriate but perhaps some rationale as to why 8 months and not 12 months or another timepoint is helpful also.
Authors: We had to adapt the intervention and assessment points to the hospital protocol. In this case, the infants are followed at 8- and 24-months corrected age. Therefore, we had to consider assessing the included infants at 8 months corrected age as the assessors were PTs from the hospital.
Regarding the ASQ-3 assessments, the PTs at the hospital were not allowed to change their protocol at 2 and 8 months corrected age, and the ASQ-3 is not a tool that is used by them regularly (they only assess infants by the AIMS). Therefore, the assessments were adapted to perform at infants’ 1 month of corrected age through phone calls by a blinded assessor.
Outcome measurement – short term vs mid term vs long term. In the discussion one of the strengths of the study refers to the ‘mid term’ follow up. Some clarification as to what is reasonably defined as ST, MT or LT follow up needs to be included as opinions will differ on this. The early assessments here are not really short term but post intervention whereas the later assessment at 8 months could be considered as short term follow up. Mid-term follow up is more likely to be considered a 2 year or preschool follow up period and longer term into school age.
Authors: Thank you for your recommendation. We have revised the manuscript and modified all the references to short- and mid-term. We have changed it into post intervention and short-term as suggested. We also have added a line in the introduction to define what is considered short, mid and long-term (lines 89-91). The following reference has also been included:
Jeong, J.; Pitchik, H.O.; Fink, G. Short-Term, Medium-Term and Long-Term Effects of Early Parenting Interventions in Low-and Middle-Income Countries: A Systematic Review. BMJ Global Health 2021, 6, doi:10.1136/bmjgh-2020-004067.
We have also deleted the paragraph related to study strength due to the mid-term follow-up.
Sources of bias:
The paragraph on limitations is good and does address the lack of pre intervention assessment and the issue of infants receiving other therapies during the study.
However there is also a significant source of bias in the inclusion criteria – only infants of parents who were in the neonatal unit for 6 hours a day were included. This is a significant selection bias. While it is important to try to have parents engaged with an intervention trial excluding those who are not always able to be in the NICU is a significant selection bias in this study and may skew data. It does also mean the data need to be interpreted in that context – that a population of infants were not included simply because parents were not able to be present as much. There is evidence that this is the group that interventions may need to target and often those infants may demonstrate a better intervention effect. The harder to engage families and the harder to follow up groups are often those that benefit greatly from intervention and education. This issue does need highlighting also.
Authors: Thank you so much for your comment. This is something that we did not consider and we totally agree with you. Therefore, we have added a short paragraph at the end of the limitations section (lines 666-672). Changes are highlighted inside the manuscript.
Conclusion:
It is more helpful if the conclusion paragraph does not have too many hypotheses and clearly summarises the finding. Suggestion for further study are appropriate.
The conclusions from this study need to be made more clearly – that there was not an effect of intervention seen on motor or global outcomes or on the parent stress. There is the possibility that higher compliance impacts parent stress positively and this could be explored with a larger study. The statement that ‘intervention could have improved infants ST fine motor, communication, problem solving etc..” is not a conclusion. The reader is left not sure if they did or did not. The results show that this was not really the case in this study and the conclusion needs to be clearly drawn.
Authors: We are grateful for your input. We have changed the conclusion as follows:
“The results of this RCT suggest that the early physiotherapy intervention was not effective on preterm infants’ motor or global outcomes nor on parental stress index. These results might be related to the dose or intensity of the intervention, but also to the poor parental compliance. Besides, there is the possibility that parental higher compliance impacted on mother’s stress index positively, as mothers with high compliance showed a lower stress index. Further research should consider these findings.“
